# A Study of the Impact of Graphene Oxide on Viral Infection Related to A549 and TC28a2 Human Cell Lines

**DOI:** 10.3390/ma14247788

**Published:** 2021-12-16

**Authors:** Piotr Kuropka, Maciej Dobrzynski, Barbara Bazanow, Dominika Stygar, Tomasz Gebarowski, Anna Leskow, Malgorzata Tarnowska, Katarzyna Szyszka, Malgorzata Malecka, Nicole Nowak, Wieslaw Strek, Rafal J. Wiglusz

**Affiliations:** 1Department of Biostructure and Animal Physiology, Wroclaw University of Environmental and Life Sciences, Kozuchowska 1, 51-631 Wroclaw, Poland; piotr.kuropka@upwr.edu.pl (P.K.); tomasz.gebarowski@upwr.edu.pl (T.G.); 2Department of Pediatric Dentistry and Preclinical Dentistry, Wroclaw Medical University, Krakowska 26, 50-425 Wroclaw, Poland; maciej.dobrzynski@umw.edu.pl; 3Department of Pathology, Wroclaw University of Environmental and Life Sciences, Norwida 31, 50-375 Wroclaw, Poland; barbara.bazanow@upwr.edu.pl; 4Department of Physiology, Faculty of Medical Sciences in Zabrze, Medical University of Silesia, Poniatowskiego 15, 40-055 Katowice, Poland; dstygar@sum.edu.pl; 5Department of Basic Sciences, Faculty of Health Sciences, Wroclaw Medical University, Grunwaldzka 2, 50-368 Wroclaw, Poland; anna.leskow@umw.edu.pl (A.L.); malgorzata.tarnowska@umw.edu.pl (M.T.); 6Institute of Low Temperature and Structure Research, Polish Academy of Sciences, Okolna 2, 50-422 Wroclaw, Poland; k.szyszka@intibs.pl (K.S.); m.malecka@intibs.pl (M.M.); n.nowak@intibs.pl (N.N.); w.strek@intibs.pl (W.S.)

**Keywords:** viral infection, Rubella virus, graphene oxide, A549 cell line, TC28a2 cell line

## Abstract

Graphene has been one of the most tested materials since its discovery in 2004. It is known for its special properties, such as electrical conductivity, elasticity and flexibility, antimicrobial effect, and high biocompatibility with many mammal cells. In medicine, the antibacterial, antiviral, and antitumor properties of graphene have been tested as intensively as its drug carrying ability. In this study, the protective effect of graphene oxide against Rubella virus infection of human lung epithelial carcinoma cells and human chondrocyte cells was examined. Cells were incubated with graphene oxide alone and in combination with the Rubella virus. The cytopathic effect in two incubation time periods was measured using DAPI dye as a percentage value of the changed cells. It was shown that the graphene oxide alone has no cytopathic effect on any of tested cell lines, while the Rubella virus alone is highly cytopathic to the cells. However, in combination with the graphene oxide percentage of the changed cells, its cytotopathicity is significantly lower. Moreover, it can be concluded that graphene oxide has protective properties against the Rubella virus infection to cells, lowering its cytopathic changes to the human cells.

## 1. Introduction

Graphene oxide (GO) is one of numerous nanomaterials whose use in various biological and medical fields seem to have great prospects. Free stable graphene oxide was discovered in 2004 by A.K. Geima and K.S. Novoselov and since then it has been the object of scientists’ interest due to its unique properties and structure [1]. Two-dimensional, one-atom layers of graphene molecules can be used to form three-dimensional structures, such as fullerenes or nanotubes. Inside them various active substances can be packed and then released in the destination site [2]. Due to the fact that pristine graphene has no electric charge—it is neither hydrophilic nor hydrophobic—it seems to be a perfect drug carrier for the storage of active substances that can be released directly into the target [3,4,5].

Currently, graphene is the subject of a number of studies in the field of drug or gene delivery [2] in antibacterial [6] or anticancer therapies [7,8]. Even though the role of this compound has already been established in human dental pulp stem cells (DPCSs) differentiation [8,9], there still remains the matter of its influence on tissues and organisms, which, according to the research of various authors, is either neutral or toxic [8,10,11,12,13].

The GO-cell response depends on the acquisition method and the ability to interact with proteins or lipids of the cell membrane [14]. Studies on the dynamics of graphene oxide absorption on animals showed the significant importance of its concentration and its proper dissolution before administration [11]. Reduced graphene-oxide nanoplatelets (rGONPs) given in doses over 100 µg/L is cytopathic and may not dissolve properly. As a result, it quickly merges into insoluble aggregates that exclude the possibility of using graphene as a nanomaterial in in vivo [15]. This is important in the case of cell cultures in which the intensity of graphene oxide absorption and its subsequent changes are most likely related to its concentration. In vitro studies also revealed that an increase in the concentration of graphene oxide in the cells induces oxidative stress [16]. Our studies showed that GO has a high affinity to the cell surface of chondrocytes, limits the cell absorption surface and the access to protein receptors for ligands.

This makes it possible to implement an antiviral strategy based on nanotechnology. In this strategy, graphene particles can be used as traps for viruses or to limit the availability of the cell surface to a virus. Although currently these possibilities apply mainly to the SARS-CoV-2 virus, they are universal mechanisms that are applicable to other viruses [17].

Interestingly, the possibility of using graphene platelets introduced into the bronchial tree was proven to cause inflammatory process [18]. It does not change the fact that graphene and multiwalled carbon nanotubes (MWCNT) in the form of single-layer meshes may serve as a kind of membrane limiting the accessibility of pneumonocytes surface for several factors, such as viruses [19].

Lots of viruses, including Rubella virus, enter the human body through the respiratory tract, depositing in the alveoli and causing inflammation that can lead to serious consequences like degradation of the alveolar walls, exudate and even neoplastic transformation. As a consequence, the cartilage present in the bronchi becomes vulnerable to the virus. Human lung epithelial carcinoma cells (A549 cell line) are wildly used as an in vitro model for type II pulmonary epithelial cells and present a rapid division index and limited secretion capacity. While pneumocytes appear to be one of the most vulnerable cells to viral infections, chondrocytes, which are a type of cells not available to various biologically active substances, are sensitive to changes of environmental conditions for example during inflammation. In addition, the chondrocytes synthesize intercellular matrix thus separating cells from the other tissues. These features mean that TC28a2 cell line, which are human chondrocyte cells established by transfecting primary cultures of costal cartilage, seems to be very promising in the establishment of a sensitive model for testing the cytopathic or cytotoxic effect of different pathogens and compounds. The cells of selected lines should not take up large amounts of GO, which, on the other hand, should adhere to the cell membranes in order to limit viral adhesion [20].

The Rubella virus (RuV) has been chosen as a model that infects most of the known human cell lines [21,22,23]. However, its affinity to the chondrocytes is still unclear. It should be noted that all types of human body cells have RuV receptors. Moreover, this receptor remains unknown [21]. The RuV is involved in many congenital diseases, such as arthritis [24], therefore it is presumed that RuV should be able to induce the cytopathic effect on chondrocytes. However, there is no up-to-date data on this topic. The RuV is particularly dangerous for the developing fetus, leading to a serious malformation [25,26]. Admittedly, the current epidemiological situation seems to be under control because of vaccinations in childhood, but an infection in pregnant women who do not have antibodies due to the earlier vaccination or disease cannot be excluded. Considering that today pro-epidemic attitudes are becoming more common, it is still important to determine whether RuV will have a cytopathic effect on chondrocytes. Furthermore, it is interesting whether graphene oxide is able to limit the spread of RuV in vitro.

Therefore, the aim of this work is the in vitro investigation of graphene oxide properties to reduce viral infection in chondrocyte and the A549 cell line caused by RuV.

## 2. Materials and Methods

### 2.1. Graphene Oxide (GO) Synthesis

All chemicals were purchased from Merck KGaA, (Darmstadt, Germany) and were used without further purification. The GO sample was prepared from natural graphite (Koh-i-noor Hardmuth, České Budějovice, Czech Republic). GO was prepared according to Zhao et al. [27] by a modified Hummers method using H_2_SO_4_, KMnO_4_ and 30% H_2_O_2_ as oxidants. In detail, 4.0 g of graphite and 3.0 g of NaNO_3_ were mixed with 300 mL of H_2_SO_4_ under stirring in an ice-water bath. Subsequently, 18.0 g of KMnO_4_ was slowly added in several portions and the mixture was then continually stirred for 5 d at room temperature. Then, the mixture was heated to 98 ± 1 °C and 560 mL of 5 wt. % H_2_O_2_ was added over about 2h, and the suspension was further stirred for 2 h at 98 ± 1 °C. The mixture was left to cool down to 60 °C and 12 mL of H_2_O_2_ (30 wt. %) was added in the suspension and further stirred for 2 h at room temperature. The obtained mixture was washed by water and underwent centrifugation at 5000 rpm several times and finally the sample was rinsed with Milli-Q water until the solution was neutral.

### 2.2. Physicochemical Characterization

A PANalyticalX’Pert Pro diffractometer (Malvern Panalytical Ltd., Malvern, UK) equipped with Ni-filtered Cu Kα radiation (λ = 0.154 nm, V = 40 kV, I = 30 mA) was employed to get powder diffraction patterns that were compared to the reference pattern from Inorganic Crystal Structure Database (ICSD).

A Philips CM-20 SuperTwin high-resolution transmission electron microscopy microscope (HRTEM) operating at 200 kV was used (Eindhoven, The Netherlands). The sample for HRTEM was prepared by dispersing a small amount of specimen in methanol and putting a droplet of the suspension on a copper microscope gird covered with carbon.

A field-emission scanning electron microscope (FEI Nova NanoSEM 230; Hillsboro, OR, USA) equipped with an EDS spectrometer (EDAX Genesis XM4) was used to determine the surface morphology. SEM images were performed at 5.0 kV in a beam deceleration mode to improve the surface sensitivity and show more detailed features of the samples.

A Thermo Scientific Nicolet iS50 FT-IR spectrometer (Waltham, MA, USA) equipped with an Automated Beamsplitter exchange system (iS50 ABX containing DLaTGSKBr detector) and an HeNe laser as an IR radiation source were used to measure IR spectra. Polycrystalline mid-IR spectra were collected in the 4000–400 cm^−1^ range in KBr pellets at the temperature of 295 K and spectral resolution of 4 cm^−1^.

### 2.3. Tissue Cultures

In the experiment, two cell cultures were used: A549 (ATCC-CCL-185TM) human lung epithelial carcinoma cells; TC28a2 (Sigma-Aldrich (St. Louis, MO, USA) SCC042)—human chondrocyte cell line purchased from ATCC (Manassas, VA, USA) and Sigma-Aldrich (St. Louis, MO, USA).

Both cell lines were cultured in Dulbecco’s modified Eagle’s medium—DMEM (Lonza; Walkersville, MD, USA) and were supplemented with 10% fetal bovine serum (FBS; Biological Industries, Israel), 4 mM/L glutamine (Biological Industries; Northern Kibbutz Beit Haemek, Israel), 100 U/mL of penicillin and 100 µg/mL of streptomycin (Sigma, Germany).

### 2.4. Cytological Investigations Tested on A549 and TC28a2 Cell Lines

The studied material was divided into 16 study groups shown in Table 1.

A549 and TC28a2 cells were incubated at 37 °C in a humidified 5% CO_2_ atmosphere in a six-well polystyrene plate (NUNC, Denmark) for 24 h.

The control group consisted of the A549 and TC28a2 cell lines, following 24 h and 48 h observation after plating (control after 24 h, control after 48 h).

The second group consisted of A549 and TC28a2 cells in 24 h and 48 h observation infected with RuV (ATCC VR-315) 30 min after plating (RuV after 24 h, RuV after 48 h).

The third group consisted of A549 and TC28a2 cells observed for 24 h and 48 h after the addition of GO, 30 min after seeding the cells (GO after 24 h, GO after 48 h). DMEM was removed and, after sonification, 1 mL of GO was dissolved in MEM (minimum essential medium; Sigma-Aldrich, Germany) as well as supplemented with 2% fetal bovine serum (FBS) and L-glutamine, concentration 68.1 µl/mL (nontoxic for normal cells), and in this state the cultures were incubated for 24 h or 48 h (37 °C, 5% CO_2_). After this time wells were washed in PBS for microscopic analysis.

The fourth group consisted of A549 and TC28a2 cells in 24 h and 48 h observation, GO was added 30 min after seeding cells (similarly as in the third group), and then after another 30 min. They were infected with RuV (GO and RuV after 24 h, GO and RuV after 48 h). Next, the cells were incubated for 24 h or 48 h (37 °C, 5% CO_2_).

After 24 h or 48 h, cells were stained with DAPI (for the visualization and analysis of cell nuclei) and Rhodamine B (for the visualization and analysis of cell cytoplasm and GO particles in it). Then the control and study groups were analyzed using a Nikon Eclipse 80i fluorescence microscope (Nikon, Amsterdam, Netherlands) equipped with an UV-2A filter (EX 330–380, DM-400, BA-420).

### 2.5. Morphometrical Analysis

The morphometric analysis of the number of cells showing cytopathic features was performed in relation to untreated cells. The measurements were made in 10 successive fields of view at a magnification of 400x.Then the obtained data were averaged and a statistical analysis was performed. The results are visualized in the graphs where percentage values are used.

### 2.6. Statistical Analysis

The statistical analysis was conducted using MS Excel 2019 (Microsoft Co.; Albuquerque, NM, USA) and Statistica 13.3 (Tibco Software Inc.; Palo Alto, CA, USA). Descriptive data was presented as a mean and a standard deviation. The distribution of the data was tested with the Shapiro–Wilk normality test. The one-way ANOVA analysis and post hoc Fisher’s least significant difference test were performed for the evaluation of differences between the tested groups. *p*-value of *p* < 0.05 was considered statistically significant.

## 3. Results

### 3.1. Structure and Morphology Analysis

The formation of the graphene oxide powders was followed by the powder XRD measurements (see Figure 1). The obtained pattern was correlated with the reference standard of the graphite that was used as a substrate, ascribed to the R-3mR space group from Inorganic Crystal Structure Database (ICSD-29123). In the case of pure graphite, a diffraction peak around 26° corresponds to the highly organized layer structure (0 0 2 plane). This diffraction peak is very small on the diffraction pattern, which is related to only to a small amount of graphite in the obtained product. Moreover, the diffraction peaks at *2θ* = 11° (0 0 1 plane) and at *2θ* = 42.7° (1 0 0 plane) were observed confirming the formation of graphene oxide. In this case, the diffraction peak derived from 0 0 1 plane is shifted towards higher value of *2θ* angle that can be related to the lower oxidation degree of graphene [28]. It was also observed that the broad and intensive diffraction peak at *2θ* = 17° is consistent with the data reported by Saladino et al. [29].

The graphene particle size and sample morphology are shown in TEM and SEM images (Figure 2 and Figure 3). The in-depth analysis of the TEM images showed that the graphene particle size distribution is very wide. The TEM images show both small ~30 nm particles and very large pieces of the sample. Contrary to the results published in [30,31], graphene oxide particles did not take the form of thin two-dimensional structures, but formed into particles with a shape difficult to define. A strong agglomeration of GO particles into very large, porous structures (SEM images) was also observed.

Figure 2 shows the TEM images with the SAED pattern as well as particle size distribution for the obtained sample. The SAED pattern shows a ring-like pattern which indicates that the examined agglomerate is a set of smaller crystalline objects with a random orientation to the electron beam (the polycrystalline nature of the sample). The diffraction pattern differs slightly from those presented in the literature for pure GO because other carbon structures may also be present in the materials, which was observed by other authors [28,29].

The infrared spectra of the obtained graphene oxides are presented in Figure 4. The FTIR spectrum of GO shows a broad peak at about 3200 cm^−1^ in the high frequency area corresponding to the stretching vibration of the -C-OH groups and water molecules adsorbed on graphene oxide. The peaks at 2920 cm^−1^ and 2853 cm^−1^ belong to the symmetric and anti-symmetric stretching vibrations of -CH_2_ groups. The peak at 1698 cm^−1^ is related to the stretching vibration of -C=O groups, at 1580 cm^−1^ with the -C-C stretching and at 1375 cm^−1^ with the -C-O stretching vibration of carboxylic acid. The peak at 1051 cm^−1^ is attributed to the -C-O bond. The presence of different kinds of oxygen functionalities in the obtained material was detected, which confirmed the formation of graphene oxide. Moreover, these polar groups, especially the surface hydroxyl groups, occur in hydrogen bonds formation between GO and water, which clarifies the hydrophilic nature of graphene oxide [28,32,33,34].

### 3.2. Cytopathic Effect Assay Results

In the cells, the morphological changes were be observed, they were followed by biochemical changes with altered gene expression regarding to the virus attachment and cell infestation. GO showed increased cytopathic effect in A549 cell lines when infected by virus after 24 h of incubation. After 48 h of incubation in both A549 and TC28a2, in the cell lines infected by RuV the level of cytopathic effect was at the similar level (Figure 5 and Figure 6).

There is an increasing trend in the occurrence of the cytopathic effect in A549 cells after the infection with RuV between 24 h and 48 h of incubation. Also, in the GO + RuV 24 h group, an increased level of infection is observed with simultaneous inhibition after 48 h. This may be due to a weak, but nevertheless existing, cytopathic effect caused by GO itself.

There is an increasing tendency of the cytopathic effect in TC28a2 cells after infection with RuV between 24 h and 48 h of incubation, but the percentage of dead cells after 48 h is much lower than in the case of A549 cells. In the GO + RuV 24 h group, in contrast to the A549 cells, fewer dead cells were observed. On the other hand, after 48 h in the GO + RuV group, a similar percentage of dead cells was observed compared to the 48 h RuV group, which indicates that GO has a protective effect at the first stage of infection (up to 24 h) and after this time its protective effect is no longer observed. The results also indicate that GO has a greater cytopathic effect on TC28a2 cells than on A549 cells.

#### 3.2.1. The A549 Cell Line—Cytopathic Changes in Cells after 24 h of Incubation

Normal A549 cells are characterized by a large, regular nucleus with a lot of loose chromatin located in the central part of the cell, in which cytoplasm is regularly distributed around the nucleus giving the cell a round shape (as shown in Figure 7a). In culture, cells grow densely and, during the division indicated by DAPI, cells can be visualized.

In our case, the cytopathic effect was related to such changes as: chromatin condensation, cytoplasm retraction and condensation around cell nucleus as well as shape of nucleus and cell.

In the A549 cell line, a different number of cells and the increased cytopathic effect (CPE) in RuV and CPE and cytotoxicity GO + RuV group were noted. The morphological changes of the cell shape with condensed cytoplasm in RuV group were observed (red arrow in Figure 7). Only few cells were normal. In the GO + RuV group, an increased content of GO (white arrow) was observed in morphologically normal cells. In the cell cytoplasm, large aggregates of GO were observed. In the GO group, most of the cells contained small droplets of GO inside the cells and some only on the surface of the cell (white arrow).

#### 3.2.2. TC28a2 Cell Line—Cytopathic Changes in Cells after 24 h of Incubation

The normal cells of the TC28a2 line show a similar structure to the cells in the A549 line, they grow loosely in the culture in contact with cytoplasmic spikes (as shown in Figure 8A).

In the graphene oxide group, a moderate uptake of GO was observed. Most of the cells were covered by the graphene oxide aggregates. However, no cytopathic effects was observed. In the RuV group, changes in cell structure were observed. Numerous cells changed their shape into the elongated or condensed sphere with reduced nucleoplasm and stained more intensively with rhodamine B. In the GO + RuV group most of the chondrocytes were covered by GO, however, no changes in cells were noted (Figure 8).

#### 3.2.3. A549 Cell Line—Cytopathic Changes in Cells after 48 h of Incubation

In the control and GO 48 groups, the number of cells is the same number and so is density. In the RuV group, only small aggregates of cells showed normal response to the dyes. In GO + RuV group, approximately half of the cells did not respond to the dye. The virus caused an increased cytopathic effect, whereas GO alone and with the virus showed reduced damage to the cells (Figure 9).

#### 3.2.4. TC28a2 Cell Line—Cytopathic Changes in Cells after 48 h of Incubation

In this group a more increased cytopathic effect in the virus group was observed. Numerous cells showed changes in their morphology. Moreover, the DAPI uptake was reduced. A similar situation in GO + RuV group was visible and numerous colonies with dividing cells were observed. It proves that the cell lines with the graphene oxide are less infected than those without the addition of it (Figure 10).

## 4. Discussion

In the presented study, the A549 and TC28a2 cell lines in relation to GO as well as Rubella were analyzed. The research on RuV was carried out intensively in the 1960s and 1970s, due to the very high frequency of this disease and numerous and serious complications it caused. However, as human immunization progressed, the disease was largely eradicated. Nowadays, however, due to the antivaccine attitudes, the number of infections may increase. For this reason, the research on RuV should be intensified, as there is no up-to-date available literature on the virus itself and on methods, including the use of nanomaterials, to limit its spread in the human population. The rubella virus enters the cell by interacting with specific receptors, which lead to the formation of an endosomal follicle. After the attachment of the lysosomes and lowering the pH inside the follicle, DNA or RNA is released from the lysosome, which leads to viral spreading over the cell and then throughout the body. The penetration of RuV induces numerous changes in living-cells which is described as cytopathic. These changes are reversible and usually not do not cause the death of living cells. Therefore, an individual cell can spread viruses throughout its life.

The studies presented here are not only aimed at checking the effect of GO on reducing the spread of RuV, but also may be useful in the case of other viruses whose infection mechanism is similar, including, for example, SARS-CoV-2.

In the presented study, changes in the A549 and TC28a2 cells phenotype and increased cytotopathic effect after RuV administration were reported. A549 cells are the most exposed cells to viruses, not only RuV, which enter the body via droplets. This means that the first and direct action of viruses and the counteraction of GO seems to be a justified choice here. On the other hand, as a result of innate RuV infections, chondrocyte cells may become virus carriers and, consequently, may lead to continuous inflammation. Despite the lack of the latest research, the authors were interested in the possibility of using GO to reduce the spread of RuV infections. Tsai et al. [35] found that graphene oxide molecules can induce apoptosis in pulmonary cells by activating the EGF-receptor. In the case of line A549, no such effect was observed. In addition, Park et al. [36] in their experiments found that graphene oxide was undetectable in lungs 24 days after administration. This was correlated with an increased cytokines level. As an inductor of inflammation, graphene allows its degradation by macrophages, nevertheless it leads to the damage to the lung parenchyma. Given this, all GO structures can be particularly toxic to lung tissue. Most likely, this process is induced by imbalance in the oxidative status of the tissues damaged by graphene oxide [37]. In this case, the main inducer of free radical formation is the damage of the mitochondria, as confirmed by Jarosz et al. [38]. The cells used in our experiment are rather secreting cells (chondrocytes) and dividing cells (A549). Therefore, their ability to uptake graphene oxide seems to be limited, thus showing less toxic effect on the studied cells. A weak cytopathic effect was observed after 24 h and 48 h in both cell lines, which was clearly stronger with chondrocytes. Bengston et al. showed the limited toxic effect of graphene oxide on mice lung cells analyzed in vitro. Despite the fact that they confirmed that GO generated free radicals, which were not cytotoxic or cytopathic to the lung cell [5].

Additionally, intensive morphological changes were detected in chondrocytes in contact with GO. However, after infection by RuV the number of cells containing GO and with normal morphology increased. In the case of A549 cells, these changes were mild and almost no changes in GO after 24 and 48 h were noted. It is interesting that, in both cell lines, the increased cytopathic effect was noted in RuV groups, whereas in presence of GO, the number of cytopathic cells was decreased in comparison to the control and GO groups. Moreover, the increased GO uptake in both cell lines in the GO groups after 24 and 48 h of observation may be caused by a viral infection, which results in changes of the cell membrane charge. Therefore, the graphene uptake increases. Similar effects were described for the Ebola virus by Gc et al. [39] where the attachment of virus to the receptor is responsible for the formation of the hexamer structure of VP40 protein at the inner membrane leaflet, which is required for virus budding and endosome formation in T lymphocyte receptors. In our case, it seems, that GO acts as a trap, which on the one hand can reduce the surface of virus attachment to the cell membrane and cover the surface receptors, and on the other hand by direct interaction it may permanently bind virus to the GO surface, thus deactivating it. Viral diseases appear to constitute an increasing proportion of human and animal diseases. Therefore, the creation of devices that would allow to effectively eliminate viruses from the environment seem to be the easiest way to achieve the effect of limiting exposure to the virus, e.g., in public places or medical facilities. In addition, there is still a need to search for various nanomaterials, including graphene-based materials that may, depending on the application, reduce viral infections by limiting their adhesion to the cell or capturing them, e.g., from flowing air [17].

## 5. Conclusions

Graphene oxide was successfully synthesized by the modified Hummers method from graphite, which was confirmed by XRD and FTIR techniques. The graphene particle size distribution is very wide from small ~30 nm particles to very large pieces. Moreover, a strong agglomeration of GO particles was observed.

It can be concluded that graphene oxide had no direct cytopathic effect on TC28a2 chondrocyte and A549 cell line. The GO has a deep impact on the uptake and reduction of viral load in vitro investigation on chosen cell lines. The obtained results may be used in a filtration mask used to prevent some viral infections. Moreover, graphene oxide showed protective properties against the Rubella virus infection to cells that can be the so-called cytopathic changes to the human cells. However, this statement requires further investigation.

## Figures and Tables

**Figure 1 materials-14-07788-f001:**
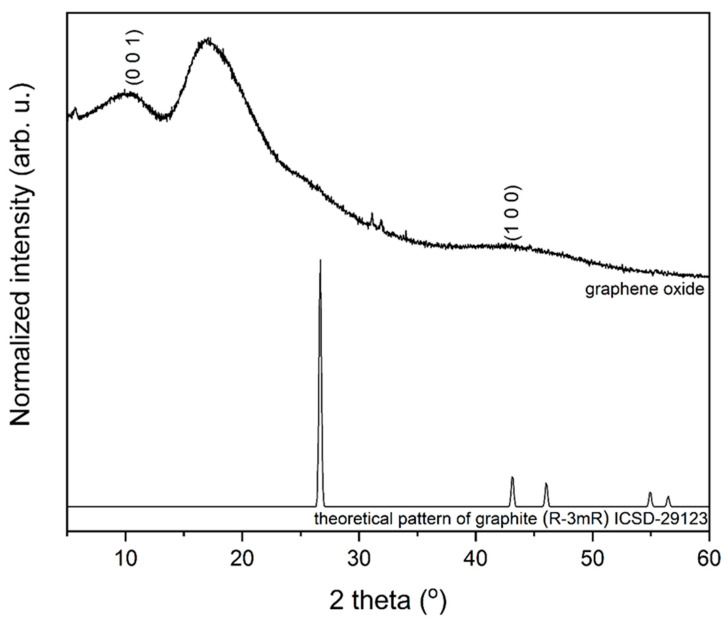
XRD patterns of obtained graphene oxide material (**above**) compared to the reference pattern of graphite (**below**).

**Figure 2 materials-14-07788-f002:**
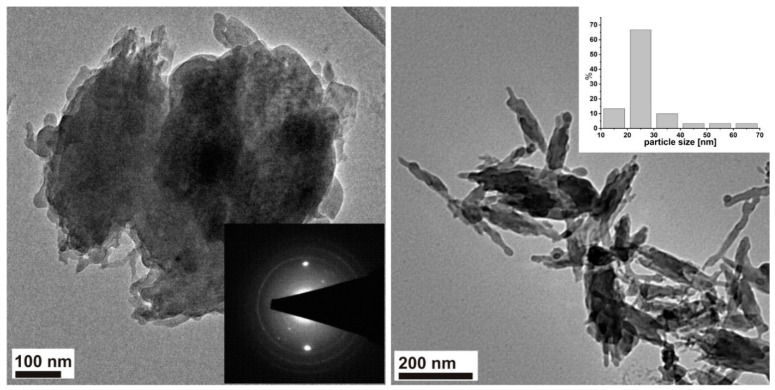
TEM images with a SAED pattern (insert in **left** image) as well as with particle size distribution (insert in **right** image) for the investigated graphene oxide material.

**Figure 3 materials-14-07788-f003:**
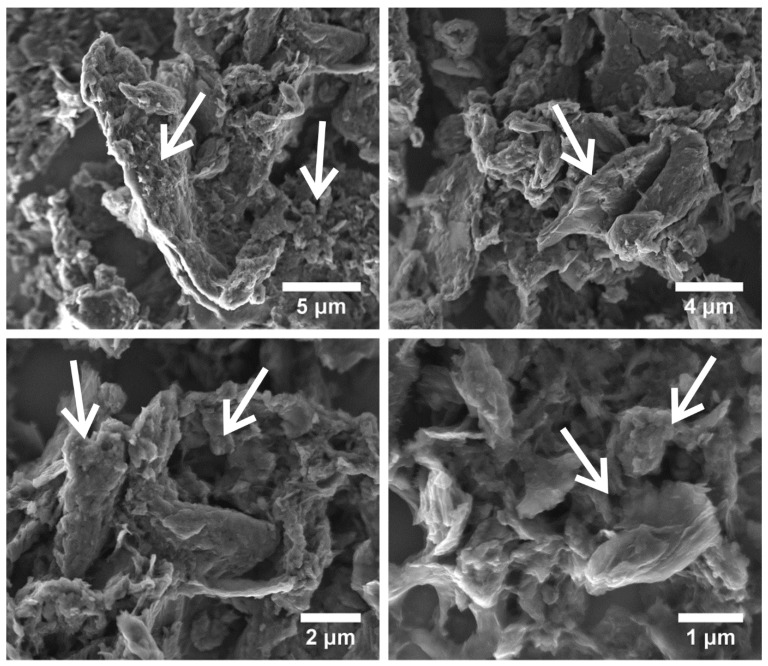
SEM images of graphene oxide material. The arrows showed particularity of obtained graphene oxide.

**Figure 4 materials-14-07788-f004:**
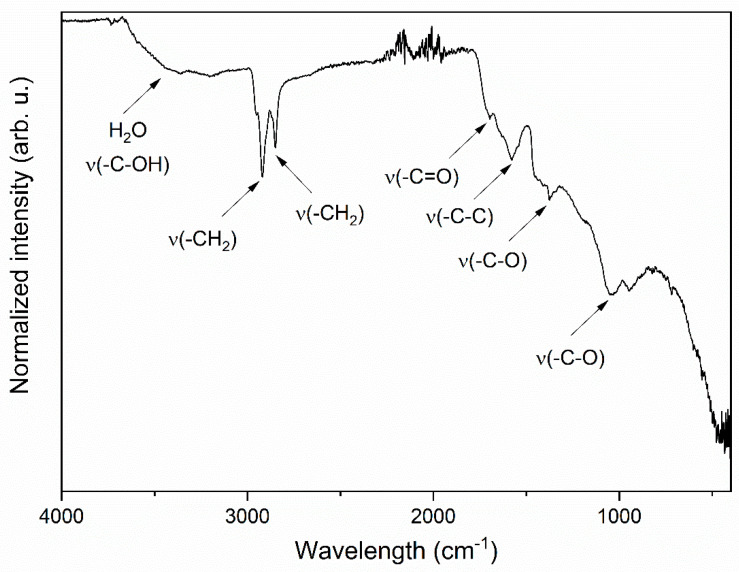
IR spectrum of graphene oxide material. The arrows showed vibration of particular functional groups.

**Figure 5 materials-14-07788-f005:**
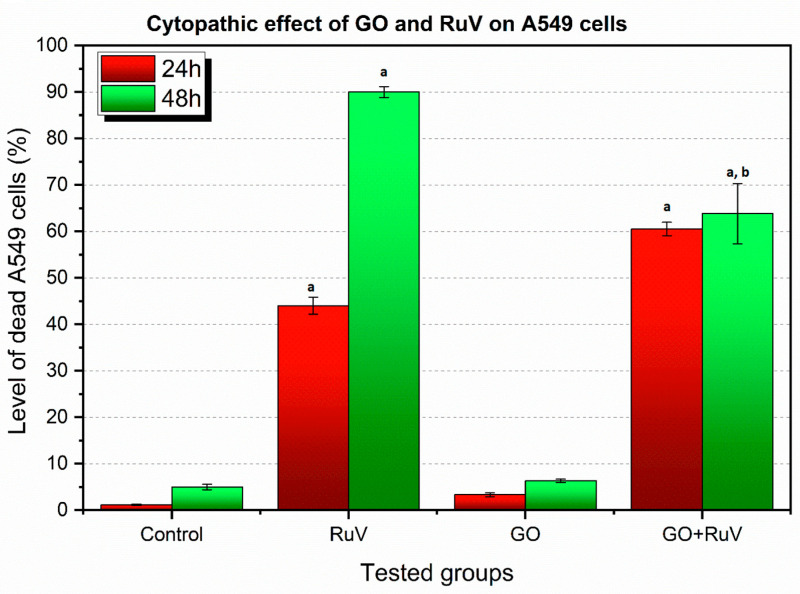
Percentage of cytopathic effect in A549 cell line. RuV—cells infected with RuV, GO—cells with graphene added, GO + RuV—cells infected with RuV and with graphene added. Statistical significance at *p* < 0.05, a: when compared to control group; b: when compared to RuV group.

**Figure 6 materials-14-07788-f006:**
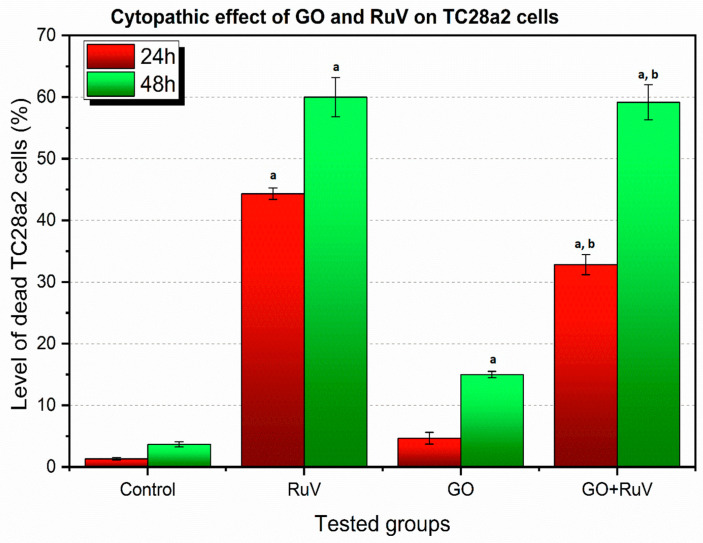
Percentage of cytopathic effect TC28a2 cell line. RuV—cells infected with RuV, GO—cells with graphene added, GO + RuV—cells infected with RuV and with graphene added. Statistical significance at *p* < 0.05—a: when compared to control group; b: when compared to RuV group.

**Figure 7 materials-14-07788-f007:**
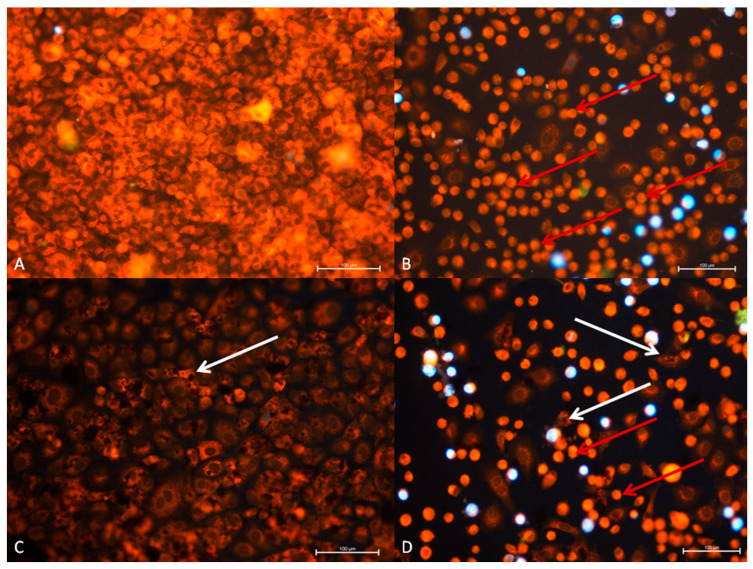
A549 (ATCC-CCL-185TM) cell line 24 h after administration of virus and GO. (**A**) Control group, (**B**) RuV group, (**C**) GO group, (**D**) RuV + GO group. Normal cells show regular, oval shape with nucleus present in the middle of the cells. Peripheral cytoplasm is well visible. In response to the RuV cytopathic effect is noted in most of the A549 cells line. Cells have reduced volume, with cytoplasm condensed around nucleus. In GO group dark aggregates are present inside and covering cells (white arrow). DAPI and Rhodamine B. Mag 100x. Scale bar 100 µm.

**Figure 8 materials-14-07788-f008:**
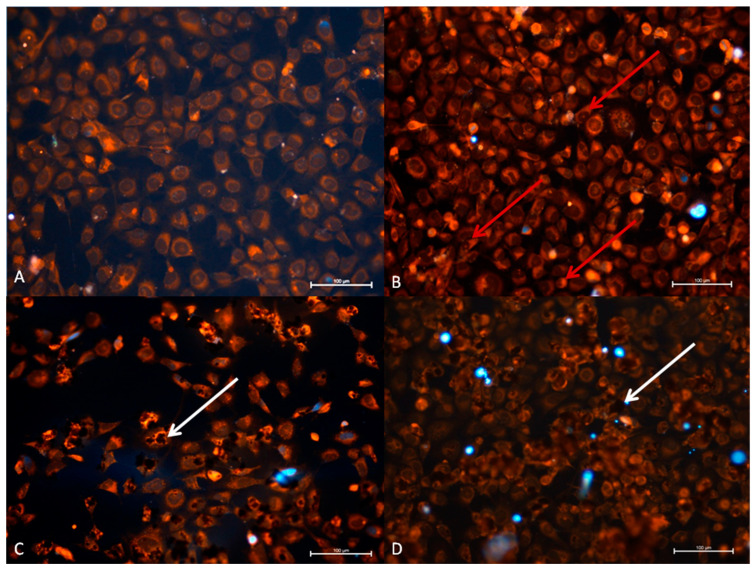
TC28a2 cell line 24 h after administration of virus and GO. (**A**) Control group, (**B**) RuV group, (**C**) GO group, (**D**) GO + RuV group. Mild cytopathic effect in cells with changed morphology (red arrow) after RuV administration is visible (red arrow).Visible graphene covering cells (white arrow) in both GO and GO + RuV treated cells. DAPI and Rhodamine B. Mag 100x. Scale bar 100 µm.

**Figure 9 materials-14-07788-f009:**
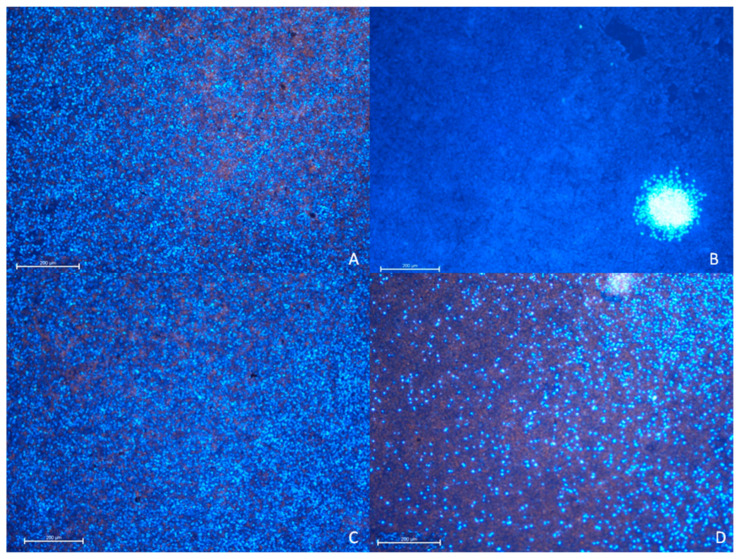
A549 (ATCC-CCL-185TM) cell line 48 h after administration of virus and GO. (**A**) Control group, (**B**) group RuV, (**C**) GO group, (**D**) GO + RuV group. Different ability of DAPI uptake (blue dots) by cells. In RuV48 group visible arrangement of surviving cell in a colony manner. Other cells showed cytopathic effect. In other GO groups, visible similar level of responding cells in comparison to the control group. In GO + RuV group about 50% of cells are not responding due to virus infestation. DAPI and Rhodamine B. Mag 40x. Scale bar 200 µm.

**Figure 10 materials-14-07788-f010:**
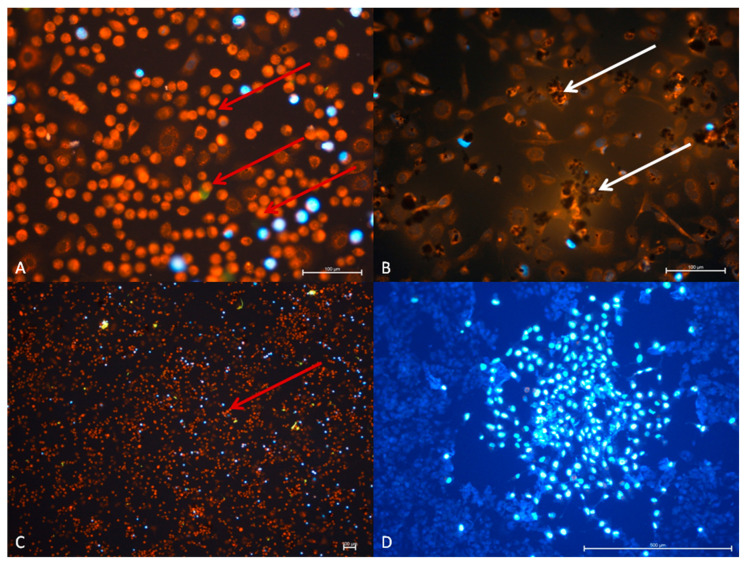
TC28a2cell line. (**A**) Group RuV, (**B**) GO group, (**C**) RuV group, (**D**) GO + RuV group. Visible changes in cell morphology after administration of virus and graphene +virus. In TC28a2 (C and D picture) cell line after 48 h after virus and graphene + virus administration increased cytopathic effect was noted. Note numerous changes in cell morphology after virus administration and reduced positive cells number in response to DAPI after 48 h (red arrow). The positive cells are located in colonies among cytopathic cells (D picture). Graphene positive cells maintain the shape of the TC28a2 cell line. DAPI + Rhodamine B. A and B—Mag 200x, C—40x, D—100x.

**Table 1 materials-14-07788-t001:** Group names according to the cell line types, GO and RuV administration and time of observation used in experiment.

Cell Line	Control after 24 h Control after 48 h	RuV after 24 h RuV after 48 h	GO after 24 h GO after 48 h	GO and RuV after 24 h GO and RuV after 48 h
A549	1/24 1/48	2/24 2/48	3/24 3/48	4/24 4/48
TC28a2	5/24 5/48	6/24 6/48	7/24 7/48	8/24 8/48

## Data Availability

Not applicable.

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
