# Peer review of "A Study of the Impact of Graphene Oxide on Viral Infection Related to A549 and TC28a2 Human Cell Lines"

_materials, 2021, doi:10.3390/ma14247788_

Round 1

Reviewer 1 Report

  1. What is reason why A549 and TC28a2 cells was chosen? The authors should provide some explanation in introduction part.
  2. There is nothing new about the Graphene oxide (GO) synthesis, what is the unique property of Go in this paper?
  3. There are some good results about Go’s protective effect towards RuV in both cell lines. However, what is the mechanism? Appropriate discussion should be provided. At the same time, the data to support the conclusion seems insufficient.

Author Response

Dear Editor,

We would like to express our sincerest gratitude to the Reviewer for the enormous efforts in criticizing the manuscript. We have considered all raised question here follows the detailed answers. Moreover, all changes we have made to the original manuscript, are marked in the red colour in the text.

 Comments and Suggestions for Authors

Q1. What is reason why A549 and TC28a2 cells was chosen? The authors should provide some explanation in introduction part.

Answer 1: It was related to the cells’ specificity. Chondrocyte cells are also inaccessible under normal and physiological conditions for viruses due to their sensitivity to external factors. In this case, we would like to check if these cells are susceptible to Rubella infection and can be protected by the graphene oxide.

The explanation was added into the manuscript.

Q2. There is nothing new about the Graphene oxide (GO) synthesis, what is the unique property of Go in this paper?

Answer 2: The synthesis or modification of the GO was not the purpose of this work. This method is well known and written in the literature.  The aim of this study was to investigate the effect of the GO on the cell protection against viruses’ ingress.

Q3. There are some good results about Go’s protective effect towards RuV in both cell lines. However, what is the mechanism? Appropriate discussion should be provided. At the same time, the data to support the conclusion seems insufficient.

 Answer 3: Corresponding changes have been made in the text.

Reviewer 2 Report

The 1.Introduction part is not provides sufficient background. Although relevant references are cited, the information is not well structured, jumps easily from one idea to another without emphasizing the importance from a scientific point of view of graphene oxide, what role they have in the biomedical area in general and in viral infections in particular and the mechanism of action in viral infections. It is very weakly emphasized (in a single sentence) the main objective of the paper, the degree of novelty and the particularities compared to other research.

            In the 2.Materials and Methods part (2.1.Graphene oxide (GO) synthesis), the synthesis method of GO is relatively well described, but it is not specified the reaction parameters changes made in the synthesis, how many samples were prepared, especially as two samples are exemplified in the measurements (sample 4, sample 5 in Figure 1). More decryption is need for part 2.3. Tissue cultures and 2.4.Cytological investigations tested on A549 and TC28a2 cell lines. What is the method, why are these investigations done, what special conditions are applied (in the discussions it refers to certain additions of substances in cell cultures that are not talked about at all in this part).

            In the 3.Results section (3.1. Structure and morphology analysis) the authors claim that they have successfully obtained the GO structures, even if in Figure 1 it is not observed at all the formation of specific picks of GO, claiming that they are small due to the fact that GO is amorphous. I consider that GOs were not obtained with the appropriate structure for the targeted application.

            In Figure 2 the author discusses the morphology of GO, where just only a small part represents GO nanoparticles, without discussing the non-uniformities that occur, the fact that large aggregates appear. This kind of aggregate nanoparticles cannot be used in biomedical applications where nanoparticles need to be relatively well dispersed.

            The experiments in the 3.2 Cytopathic effect assay results (Figure 5, Figure 6) need to be more clearly describe, underline the role of each used component. Someone unfamiliar with this kind of study will not understand the differences between the obtained percentages if they are not explained.

            In 3.2.1. 24 h of incubation. The A549 cell line, 24 h of incubation. TC28a2 cell line, After 48 hours. A549cell line, After 48 hours. TC28a2 cell line (Figure 6,7,8,9,10) some optical microscope images are presented (I presume, because it does not speak anywhere about the measurement mode) in which the changes in morphology of the cells are presented in the presence and absence of GO and/or the viruses. The author need to give a theoretical description of what these cells should look like and only after, discus the induced changes. It is not at all clear what they tried to show by the added white arrows, if the structure and initial potency are not known.

            The 4. Discussion part is to short and not concluding. What important results were obtained? The as obtained results are not discussed in detail, what novelties have been obtained compared to other researches. The author talks about cell modification, but without saying why these cells present some morphological changes. Is not described which is the GO mechanism of action at the cellular level. Personally, I think the results are very poor.

            The 5. Conclusion section is poor and not concluding.

Author Response

Dear Editor,

We would like to express our sincerest gratitude to the Reviewer for the enormous efforts in criticizing the manuscript. We have considered all raised question here follows the detailed answers. Moreover, all changes we have made to the original manuscript, are marked in the red colour in the text.

Comments and Suggestions for Authors

Q1. The 1. Introduction part is not provides sufficient background. Although relevant references are cited, the information is not well structured, jumps easily from one idea to another without emphasizing the importance from a scientific point of view of graphene oxide, what role they have in the biomedical area in general and in viral infections in particular and the mechanism of action in viral infections. It is very weakly emphasized (in a single sentence) the main objective of the paper, the degree of novelty and the particularities compared to other research.

Answer 1: The changes have been made in the introduction .

Q2. In the 2. Materials and Methods part (2.1. Graphene oxide (GO) synthesis), the synthesis method of GO is relatively well described, but it is not specified the reaction parameters changes made in the synthesis, how many samples were prepared, especially as two samples are exemplified in the measurements (sample 4, sample 5 in Figure 1). More decryption is need for part 2.3. Tissue cultures and 2.4. Cytological investigations tested on A549 and TC28a2 cell lines. What is the method, why are these investigations done, what special conditions are applied (in the discussions it refers to certain additions of substances in cell cultures that are not talked about at all in this part).

Answer 2: The changes have been added to 2.3 and 2.4 sections.

Q3. In the 3.Results section (3.1. Structure and morphology analysis) the authors claim that they have successfully obtained the GO structures, even if in Figure 1 it is not observed at all the formation of specific picks of GO, claiming that they are small due to the fact that GO is amorphous. I consider that GOs were not obtained with the appropriate structure for the targeted application.

Answer 3: We observed the broad and intensive diffraction peak at 2theta = 11o (001 plane) and broad and weak peak at 2theta = 42.7o (100 plane) characteristic for graphene oxide confirming its formation. In our case, diffraction peak derived from 001 plane is shifted towards higher value of 2theta that can be related to lower oxidation degree of graphene oxide that was shown by K. Krishnamoorthy et. (Carbon 53 (2013) 38-49; https://doi.org/10.1016/j.carbon.2012.10.013). We observed also the broad and intensive diffraction peak at 2theta = 17o what is consistent with the data reported by M.Lu. Saladino et. al. (Materials 13 (2020) 1980; doi:10.3390/ma13081980). The changes have been made and added in the manuscript.

Q4. In Figure 2 the author discusses the morphology of GO, where just only a small part represents GO nanoparticles, without discussing the non-uniformities that occur, the fact that large aggregates appear. This kind of aggregate nanoparticles cannot be used in biomedical applications where nanoparticles need to be relatively well dispersed.

Answer 4: In the case of the nanosized materials there are observed the agglomerated particles, especially in the water dispersion. This effect is related to specific surface area of nanostructured materials. It should not interfere with medical applications because it is not a dominant effect of the GO material.

Q5. The experiments in the 3.2 Cytopathic effect assay results (Figure 5, Figure 6) need to be more clearly describe, underline the role of each used component. Someone unfamiliar with this kind of study will not understand the differences between the obtained percentages if they are not explained.

Answer 5: Changes have been made to the text.

Q6. In 3.2.1. 24 h of incubation. The A549 cell line, 24 h of incubation. TC28a2 cell line, After 48 hours. A549cell line, After 48 hours. TC28a2 cell line (Figure 6,7,8,9,10) some optical microscope images are presented (I presume, because it does not speak anywhere about the measurement mode) in which the changes in morphology of the cells are presented in the presence and absence of GO and/or the viruses. The authors need to give a theoretical description of what these cells should look like and only after, discus the induced changes. It is not at all clear what they tried to show by the added white arrows, if the structure and initial potency are not known.

Answer 6: The changes have been made in the result section.

Q7. The 4. Discussion part is to short and not concluding. What important results were obtained? The as obtained results are not discussed in detail, what novelties have been obtained compared to other researches. The author talks about cell modification, but without saying why these cells present some morphological changes. Is not described which is the GO mechanism of action at the cellular level. Personally, I think the results are very poor.

Answer 7: The changes have been made in the discussion section.

Q8. The 5. Conclusion section is poor and not concluding.

Answer 8: The changes have been made in the conclusion section.

Reviewer 3 Report

The authors have chosen a particularly interesting and relevant topic and have a good beginning of a nice manuscript that will interest many readers of the journal. In the manuscript entitled "A study of the graphene oxide impact on viral infection related to A549 and TC28a2 human cell lines", the authors present their work using graphene oxide to prevent viral attachment in two different cell lines. 

At this time, however, the current manuscript is not ready for publication. The experimental design is sound, but the conclusions are muddled by poor reporting and lack of a statistical analysis. The most significant issues are as follows:

  • The authors are recommended to use an English editing service, or at least consult with a native English speaker.
  • The rationale for using the two cell lines is not made clear until line 254. These types of details need to be made clearer for the reader much earlier in the text.
  • The manuscript is riddled with contradictions. For example,
    • In the introduction, the authors are clearly trying to demonstrate the variability and large abundance of unknowns in the field of graphene oxide research. More explanation is needed to demonstrate how the authors are sifting through and/or utilizing this previous knowledge.
    • The authors express surprise at the structure of the graphene oxide synthesized in the results section, but then claim a successful synthesis in the discussion section.
    • In the conclusions section, the authors final thoughts are to use the graphene oxide for a completely different purpose than what they have been studying in their experiments. While extending the work to other fields is an appropriate topic to broach, it is not appropriate as a final summary of the work presented.
  • The materials and methods section is incomplete and not properly organized.  In particular,
    • One of the cell line names is missing in line 97, but that entire sentence is irrelevant for the section in which it is contained. i.e., the cell lines used do not pertain to the synthesis of graphene oxide.
    • Table 1 is rather irrelevant. Textually explaining the various groups would be more beneficial than forcing readers to remember random strings of numbers.
    • The comment concerning nontoxicity of GO in line 135 is not appropriate for the methods section, but it is valuable information that belongs in the results section!
  • No statistics were performed.

Author Response

Dear Editor,

We would like to express our sincerest gratitude to the Reviewer for the enormous efforts in criticizing the manuscript. We have considered all raised question here follows the detailed answers. Moreover, all changes we have made to the original manuscript, are marked in the red colour in the text.

Q1. The authors are recommended to use an English editing service, or at least consult with a native English speaker.

Answer 1: The manuscript has been checked by a native speaker.

Q2. The rationale for using the two cell lines is not made clear until line 254. These types of details need to be made clearer for the reader much earlier in the text.

Answer 2: A full justification for the choice of the cell lines has been shown in the Introduction

The manuscript is riddled with contradictions. For example:

Q3. In the introduction, the authors are clearly trying to demonstrate the variability and large abundance of unknowns in the field of graphene oxide research. More explanation is needed to demonstrate how the authors are sifting through and/or utilizing this previous knowledge.

Answer 4: The changes have been made in the text.

Q4. The authors express surprise at the structure of the graphene oxide synthesized in the results section, but then claim a successful synthesis in the discussion section.

Answer 4: The graphene oxide was synthesised successfully what was confirmed by different measurements techniques.

Q5. In the conclusions section, the authors final thoughts are to use the graphene oxide for a completely different purpose than what they have been studying in their experiments. While extending the work to other fields is an appropriate topic to broach, it is not appropriate as a final summary of the work presented.

Answer 5: The changes have been made in the text

The materials and methods section is incomplete and not properly organized. In particular:

Q6. One of the cell line names is missing in line 97, but that entire sentence is irrelevant for the section in which it is contained. i.e., the cell lines used do not pertain to the synthesis of graphene oxide.

Answer6:Changes have been made in the text. We have reorganized Materials and Method section and deleted the information about cell lines from graphene oxide synthesis section.

Q7. Table 1 is rather irrelevant. Textually explaining the various groups would be more beneficial than forcing readers to remember random strings of numbers.

Answer 7: In our opinion, the table shows the data and can be used in the other publications as a representation of the studied groups.

Q8. The comment concerning nontoxicity of GO in line 135 is not appropriate for the methods section, but it is valuable information that belongs in the results section!

 Answer 8: The changes have been made in the text and information about nontoxicity have been transferred to the results section from Materials and methods section.

Q9. No statistics were performed.

Answer 9: A statistics’ section has been addend into the manuscript. The obtained GO showed cytotoxicity at lower concentration i.e., 1:160 (GO: MEM) therefore, it cannot be concluded that it is non-toxic, but the cytotoxic effect was not observed at the used concentration.

Reviewer 4 Report

Reviewer’s comments:

The manuscript entitled ‘A study of the graphene oxide impact on viral infection related to A549 and TC28a2 human cell lines’ has been peer-reviewed. The manuscript demonstrates the protective role of graphene oxide against viral infection through in vitro biological characterization. We have provided the following comments to improve the manuscript.

Major concerns:

1) Graphene is one of the most tested material since its discovery in 2004. It is known for its specific properties in example electrical conductivity, elasticity and flexibility, antimicrobial effect, and high biocompatibility with many mammals’ cells. In medicine antibacterial, antiviral, and anti-tumor properties of graphene are tested as intensively as its drug carrying ability.

In the abstract, the author has provided a history of graphene instead of the aim of the work. Please modify the abstract.

2) Due to the fact that graphene oxide has no electric charge - it is neither hydrophilic nor hydrophobic - it seems to be a perfect carrier of drugs or a storage of active substances that can be released directly into the target [3–5].

Graphene possesses hydrophilic nature exhibiting oxygen-containing functional groups. Hence, the above statement may be wrong.

3) Cells of selected lines should not absorb large amounts of GO, which on the other hand, should adhere to the cell membranes in order to limit viral adhesion [17].

The author should be aware of the usage of technical words. The term ‘absorb’ is not suitable for this statement.

4) FTIR spectrum of graphene oxide should be properly explained. (Page 6, lines 167-171). Even in the figure, similar bonds (-C-O, -CH2) have been displayed without any difference. A distinct difference between similar bonds should be properly mentioned.

5) Figures 7, 8, and 10. How the author confirms and indicates graphene oxide by the white arrow? How is the size of GO similar to that of cells treated? What do white color spheres mean? Mention the white spheres in figure legends.

Author Response

Dear Editor,

We would like to express our sincerest gratitude to the Reviewer for the enormous efforts in criticizing the manuscript. We have taken into account all raised question here follows the detailed answers. Moreover, all changes we have made to the original manuscript, are marked in the red colour in the text.

Comments and Suggestions for Authors

The manuscript entitled ‘A study of the graphene oxide impact on viral infection related to A549 and TC28a2 human cell lines’ has been peer-reviewed. The manuscript demonstrates the protective role of graphene oxide against viral infection through in vitro biological characterization. We have provided the following comments to improve the manuscript.

Major concerns:

Q1. Graphene is one of the most tested material since its discovery in 2004. It is known for its specific properties in example electrical conductivity, elasticity and flexibility, antimicrobial effect, and high biocompatibility with many mammals’ cells. In medicine antibacterial, antiviral, and anti-tumor properties of graphene are tested as intensively as its drug carrying ability.

Answer 1: The graphene has been intensively studied for many years that absolutely has not excluded the need for further research on this material in different forms – graphene oxide. Our research has focused on graphene oxide and its protective effect against cells that were exposed to viruses.

 Q2. In the abstract, the author has provided a history of graphene instead of the aim of the work. Please modify the abstract.

Answer 2: The abstract has been modified.

Q3. Due to the fact that graphene oxide has no electric charge - it is neither hydrophilic nor hydrophobic - it seems to be a perfect carrier of drugs or a storage of active substances that can be released directly into the target [3–5].

Graphene possesses hydrophilic nature exhibiting oxygen-containing functional groups. Hence, the above statement may be wrong.

Answer 3: It should be the graphene oxide but not the graphene. Therefore, it is necessary to distinguish graphene from graphene oxide. An error has been made in our manuscript and we would like to thank for your attention. It has been changed in the manuscript.

Q4. Cells of selected lines should not absorb large amounts of GO, which on the other hand, should adhere to the cell membranes in order to limit viral adhesion [17].

The author should be aware of the usage of technical words. The term ‘absorb’ is not suitable for this statement.

Answer4: It has been changed the "absorb" word to the "uptake" in the manuscript.

Q4. FTIR spectrum of graphene oxide should be properly explained. (Page 6, lines 167-171). Even in the figure, similar bonds (-C-O, -CH2) have been displayed without any difference. A distinct difference between similar bonds should be properly mentioned.

Answer 4: There are explained differences between -C-O and -CH2 bonds (please see lines 181-192).

Q5. Figures 7, 8, and 10. How the author confirms and indicates graphene oxide by the white arrow? How is the size of GO similar to that of cells treated? What do white colour spheres mean? Mention the white spheres in figure legends.

Answer 5: The graphene oxide generates different types of coatings that cover the cells. Moreover, the cells’ fluorescence has been not observed. This has been also added to the picture description.

Reviewer 5 Report

Here in attachment you will find my considerations

Author Response

Dear Editor,

We would like to express our sincerest gratitude to the Reviewer for the enormous efforts in criticizing the manuscript. We have considered all raised question here follows the detailed answers. Moreover, all changes we have made to the original manuscript, are marked in the red colour in the text.

Comments and Suggestions for Authors

The ms: A study of the graphene oxide impact on viral infection related to A549 and TC28a2 human cell lines is well organized and interesting ,but needs some revision.

Q1.1 Page 2, line 6: it should be added that: even though a role for this compound has been stated in DPCSs differentiation” adding the references, reported:

  • Graphene oxide enrichment of collagen membranes improves DPSCs differentiation and controls inflammation occurrence.Radunovic M, De Colli M, De Marco P, Di Nisio C, Fontana A, Piattelli A, Cataldi A, Zara S.J Biomed Mater Res A. 2017 Aug;105(8):2312-2320. doi: 10.1002/jbm.a.36085. Epub 2017 May 30.
  • Osteoblastic Differentiation on Graphene Oxide-Functionalized Titanium Surfaces: An In Vitro Study. Di Carlo R, Di Crescenzo A, Pilato S, Ventrella A, Piattelli A, Recinella L, Chiavaroli A, Giordani S, Baldrighi M, Camisasca A, Zavan B, Falconi M, Cataldi A, Fontana A, Zara S.Nanomaterials (Basel). 2020 Apr 1;10(4):654. doi: 10.3390/nano10040654.

Answer1: The changes have been added

Q2.MM section: Scanning electron Microscopy analysis method should be included

Answer2: The method of scanning electron microscopy analysis has been added in the manuscript (please see section Materials and Methods line 114 – 118).

Q3. MM section Cytological investigation…..:study groups should be described

Answer 3: The MM Section has been changed.

Q4. Statistical analysis methods used completely lack in this study

Answer 4: The MM 4 section has been added.

 Q5. Results: arrows should be added to SEM pictures to better indicate graphene oxide peculiarities

Answer5: According to reviewer suggestion, the arrows have been added in Figure 3.

 Q6.I think is not correct to quantify in % the cytopathic effect of GO on A549 and TC28a2 cells, but its cytotoxic effect, acquired for example by MTT analysis or LDH assay, should be added to show, the % of dead cells. , since in the ms that:cells are in the same number and density” is reported “page 9 ,line 1

Answer 6: It was shown that cell mortality was at a low level because the goal of the study was not to investigate the cytotoxicity of the cells (their mortality in the given environment) but cytopathicity, i.e., changes in cell morphology under the influence of the external environment. Rubella virus is cytopathic, not cytotoxic.

Q7. In addition the cytopathic effect should be better explained by TEM analysis, that,at the same time,could evidence the graphene oxide uptake by the cells, as reported in discussion (Page 12,line12)

Answer 7: It has been decided that TEM images would not be measured in this study due to the dynamic changes in cells. It was continued  at the electron microscopy analysis.

Round 2

Reviewer 1 Report

it can be accepted

Author Response

Dear Editor,

We would like to express our sincerest gratitude to the Reviewer for the enormous efforts in criticizing the manuscript.

Comments and Suggestions for Authors:

It can be accepted

Response 1: Thank you very much once again for your response and suggestions.

Reviewer 2 Report

Although some additional phrases have been introduced in the Introduction part to clarify the importance from a scientific point of view of grapheme oxide, the purpose of the article is still very weakly underlined in a single sentence (“Therefore, the aim of this work is in vitro investigation of graphene oxide properties to reduce viral infection in chondrocyte and A549 cell line caused by RuV. I am worried about the novelty of this paper compared to other research.

In the 2. Materials and Methods part (2.1. Graphene oxide (GO) synthesis I asked the authors to specify the changes in the reaction parameters, so as to justify the two samples exemplified in the measurements (sample 4, sample 5 in Figure 1)

The authors did not provide any information on this direction.

I asked for more decryption for part 2.3. Tissue cultures and 2.4. Cytological investigations tested on A549 and TC28a2 cell lines. (What is the method, why are these investigations done, what special conditions are applied (in the discussions it refers to certain additions of substances in cell cultures that are not talked about at all in this part).

The authors did not provide any information on this direction.

The authors did not make any changes in this regard, although they state that the changes were made.

Graphene particle size and sample morphology are shown in TEM and SEM images (Figure 2 and 3). In-depth analysis of the TEM images showed that the graphene particle size distribution is very wide. The TEM images show both small ~ 30nm particles and very large pieces of the sample.

Please provide a size distribution graph determined from the TEM images to support the size provided.

The diffraction pattern differs slightly from those presented in the literature for pure GO because the sample contains residues of various reagents used during the synthesis (crystalline)”

A sample prepared correctly, should not contain residues, and if it does contain some purification methods of the sample should be applied before measurements.

In Figure 2 the author discusses the morphology of GO, where just only a small part represents GO nanoparticles, without discussing the non-uniformities that occur, the fact that large aggregates appear. This kind of aggregate nanoparticles cannot be used in biomedical applications where nanoparticles need to be relatively well dispersed.

The author answer that “In the case of the nanosized materials there are observed the agglomerated particles, especially in the water dispersion. This effect is related to specific surface area of nanostructured materials. It should not interfere with medical applications because it is not a dominant effect of the GO material

I consider that for biomedical application the size and a relatively uniform distribution of the nanoparticles are necessary. To avoid the nanoparticles aggregation different surface functionalization are made.

For the experiments in the 3.2 Cytopathic effect assay results (Figure 5, Figure 6) I asked the authors to be more clearly describe, underline the role of each used component. Someone unfamiliar with this kind of study will not understand the differences between the obtained percentages if they are not explained.

The authors did not provide any more explanation on this direction.

In 3.2.1. 24 h of incubation. The A549 cell line, 24 h of incubation. TC28a2 cell line, After 48 hours. A549cell line, After 48 hours. TC28a2 cell line (Figure 6,7,8,9,10) some optical microscope images are presented (I presume, because it does not speak anywhere about the measurement mode) in which the changes in morphology of the cells are presented in the presence and absence of GO and/or the viruses. The authors need to give a theoretical description of what these cells should look like and only after, discus the induced changes. It is not at all clear what they tried to show by the added white arrows, if the structure and initial potency are not known.

Not enough explanations were provided.

The 4. Discussion part is to short and not concluding. What important results were obtained? The as obtained results are not discussed in detail, what novelties have been obtained compared to other researches. The author talks about cell modification, but without saying why these cells present some morphological changes. Is not described which is the GO mechanism of action at the cellular level. Personally, I think the results are very poor.

Not enough explanations were provided.

  1. Conclusion section is poor and not concluding.

This part has not been improved

Author Response

Dear Editor,

We would like to express our sincerest gratitude to the Reviewer for the enormous efforts in criticizing the manuscript. We have considered all raised question here follows the detailed answers. Moreover, all changes we have made to the original manuscript, are marked in the red color in the text.

Comments and Suggestions for Authors:

Q1. Although some additional phrases have been introduced in the Introduction part to clarify the importance from a scientific point of view of grapheme oxide, the purpose of the article is still very weakly underlined in a single sentence (“Therefore, the aim of this work is in vitro investigation of graphene oxide properties to reduce viral infection in chondrocyte and A549 cell line caused by RuV. I am worried about the novelty of this paper compared to other research.

Response 1: The last research on this subject was carried out in the 1970s. Single studies on rabbits, further observations on other animal material, appeared in the 2004. This was due, inter alia, to the use of vaccines, which significantly reduced the incidence of RuV infections in humans. However, as mentioned in the text, anti-vaccine attitudes have re-increased congenital RuV infections and complications in pregnant women. This is particularly true for maternal arthritis, as well as for congenital developmental disorders in children, including heart problems, loss of hearing and eyesight, intellectual disability, and liver or spleen damage. This suggests that viruses can be transmitted vertically and viruses can survive in places that are inaccessible to the immune system, such as cartilage tissue. In our work, we used RuV as a model virus whose spread may be limited by graphene oxide. 

Q2. In the 2. Materials and Methods part (2.1. Graphene oxide (GO) synthesis I asked the authors to specify the changes in the reaction parameters, so as to justify the two samples exemplified in the measurements (sample 4, sample 5 in Figure 1)

The authors did not provide any information on this direction.

Response 2: Thank you for paying attention again to XRD measurement where two samples were shown. We have been corrected the Figure 1 and only one material has been presented. This error is due to the fact that during synthesis, there were obtained two samples but the experiments were conducted only with one sample.

Q3. I asked for more decryption for part 2.3. Tissue cultures and 2.4. Cytological investigations tested on A549 and TC28a2 cell lines. (What is the method, why are these investigations done, what special conditions are applied (in the discussions it refers to certain additions of substances in cell cultures that are not talked about at all in this part).

The authors did not provide any information on this direction.

The authors did not make any changes in this regard, although they state that the changes were made.

Response 3: We completely rearranged sections 2.3 and 2.4 and completed the missing information regarding our cell culture and experiment. We have also completed the descriptions of individual groups.

Q4.Graphene particle size and sample morphology are shown in TEM and SEM images (Figure 2 and 3). In-depth analysis of the TEM images showed that the graphene particle size distribution is very wide. The TEM images show both small ~ 30nm particles and very large pieces of the sample.

Please provide a size distribution graph determined from the TEM images to support the size provided.

Response 4: The particle size distribution of obtained graphene oxide has been added into the manuscript.

Q5.The diffraction pattern differs slightly from those presented in the literature for pure GO because the sample contains residues of various reagents used during the synthesis (crystalline)”

A sample prepared correctly, should not contain residues, and if it does contain some purification methods of the sample should be applied before measurements.

Response 5: Thank you for your remark. The material was purified and does not contain any residues of used chemicals, so the confusing part of sentence has been removed.

Q6. In Figure 2 the author discusses the morphology of GO, where just only a small part represents GO nanoparticles, without discussing the non-uniformities that occur, the fact that large aggregates appear. This kind of aggregate nanoparticles cannot be used in biomedical applications where nanoparticles need to be relatively well dispersed.

Response 6: Before using this material in biological experiment, graphene oxide was sonicated to break up particle agglomerates and well disperse them in water.

Q7. The author answer that “In the case of the nanosized materials there are observed the agglomerated particles, especially in the water dispersion. This effect is related to specific surface area of nanostructured materials. It should not interfere with medical applications because it is not a dominant effect of the GO material

I consider that for biomedical application the size and a relatively uniform distribution of the nanoparticles are necessary. To avoid the nanoparticles aggregation different surface functionalization are made.

Response 7: We agree with the Reviewer that the uniform distribution of the nanosized material is necessary. In the case of the GO material that was obtained with the help of  a modified Hummers method, the surface of the GO material is  modified with different functional groups like: SO42–, MnO4 or NO3 that reduce the agglomeration in water dispersion.

Q8. For the experiments in the 3.2 Cytopathic effect assay results (Figure 5, Figure 6) I asked the authors to be more clearly describe, underline the role of each used component. Someone unfamiliar with this kind of study will not understand the differences between the obtained percentages if they are not explained.

The authors did not provide any more explanation on this direction.

Response 8: We have added descriptions to section 3.2 to better explain the results in the graphs. We have added a description of the effects of GO and RuV on the individual cell lines tested and the differences in these effects.

Q9. In 3.2.1. 24 h of incubation. The A549 cell line, 24 h of incubation. TC28a2 cell line, After 48 hours. A549cell line, After 48 hours. TC28a2 cell line (Figure 6,7,8,9,10) some optical microscope images are presented (I presume, because it does not speak anywhere about the measurement mode) in which the changes in morphology of the cells are presented in the presence and absence of GO and/or the viruses. The authors need to give a theoretical description of what these cells should look like and only after, discus the induced changes. It is not at all clear what they tried to show by the added white arrows, if the structure and initial potency are not known.

Not enough explanations were provided.

Response 9: To better illustrate the differences between the groups and the normal state, we have added an additional description of normal cells (which are in Figures 7A and 8A, as normal cells are the control). 

Q10. The 4. Discussion part is to short and not concluding. What important results were obtained? The as obtained results are not discussed in detail, what novelties have been obtained compared to other research. The author talks about cell modification, but without saying why these cells present some morphological changes. Is not described which is the GO mechanism of action at the cellular level. Personally, I think the results are very poor.

Not enough explanations were provided.

Response 10: We added information about the results we obtained, their relevance for science and human life. GO mechanism of action at the cellular level was not the goal of the research, the results of which were presented in the reviewed work. This topic is of course interesting and requires further analysis that we plan to do in the future.

Q11. The 5. Conclusion section is poor and not concluding.

This part has not been improved

Response 11: We have rearranged the conclusion section.

Reviewer 3 Report

After revision, the manuscript has not been adequately improved to merit publication.  In particular:

  • The use of the English language is consistently poor.  Grammar and syntax errors fill the manuscript.
  • Some statistics and minor revisions have been performed.  These improvements do help the manuscript, but poor reporting negates their impact:
    • Figures 1-4 have titles but no descriptive legends.  Legends are available on the remaining figures, but they lack the detail and clarity to properly describe the images.
    • Statistics were only performed on one experiment.  The authors claim that differences were “proved” for experiments that lack statistical proof.  (e.g., lines 271-275)

Overall, the work performed is important and some interesting results seem to be obtained.  The authors, however, need to perform more comprehensive statistical analyses on all of their data rather than claiming significance on qualitative observations.  At this time, the work that has been performed is not ready for publication.

Author Response

Dear Editor,

We would like to express our sincerest gratitude to the Reviewer for the enormous efforts in criticizing the manuscript. We have considered all raised question here follows the detailed answers. Moreover, all changes we have made to the original manuscript, are marked in the red color in the text.

Comments and Suggestions for Authors:

Q1. After revision, the manuscript has not been adequately improved to merit publication. In particular:

  • The use of the English language is consistently poor.  Grammar and syntax errors fill the manuscript.

Response 1: We have checked and corrected English language with a native speaker.

Q2. Some statistics and minor revisions have been performed.  These improvements do help the manuscript, but poor reporting negates their impact:

    • Figures 1-4 have titles but no descriptive legends.  Legends are available on the remaining figures, but they lack the detail and clarity to properly describe the images.

Response 2: The 1-4 figures description has been extended.

Q3. Statistics were only performed on one experiment.  The authors claim that differences were “proved” for experiments that lack statistical proof.  (e.g., lines 271-275)

Overall, the work performed is important and some interesting results seem to be obtained.  The authors, however, need to perform more comprehensive statistical analyses on all of their data rather than claiming significance on qualitative observations.  At this time, the work that has been performed is not ready for publication.

Response 3: The entire “Material and Methods” and “Results” sections have been revised, along with a description of how statistical data were collected and processed.

Reviewer 4 Report

The authors have improved the manuscript by addressing the reviewer's questions. Hence, the manuscript can be accepted in its present form.

Author Response

Dear Editor,

We would like to express our sincerest gratitude to the Reviewer for the enormous efforts in criticizing the manuscript.

Comments and Suggestions for Authors:

The authors have improved the manuscript by addressing the reviewer's questions. Hence, the manuscript can be accepted in its present form.

Response 1: Thank you very much once again for your response and suggestions.
